# Replication study of "Privacy-preserving Collaborative Learning with Automatic Transformation Search"

## Reproducibility Summary

**Scope of Reproducibility**

We evaluate the reproducibility of this paper, which proposes an automatic search algorithm to find privacy preserving transformation policies in the setting of federated learning. To achieve this we test all the main claims made by the authors by rerunning the experiments and reporting the reproduced results. We further extend their work to a new dataset.

**Methodology**

We perform all experiments using the model architectures and hyperparameters proposed by the authors. We use the same datasets and extend their work to include one new dataset. A codebase was available which enables us to reproduce some of the results. However we deliver a contribution by fully re-implementing the codebase in PyTorch Lightning to ensure all components are modular, and experiments can be easily executed and extended, to the benefit of future research using the authors' method. All experiments are performed on Nvidia GTX 1080 GPUs.

**Results**

Overall we find the same results as the authors: searched transformation policies can defend users in federated learning from reconstruction attacks. These transformations also have negligible impact on training efficiency and model accuracy. However we do not observe the reported correlation between the authors privacy-score and PSNR. We are in contact with the authors about this. Also we find that the results differ greatly from image to image, with standard deviations in PSNR values of over $25\%$ the value. This means that for some specific images the method is not effective.

**What was easy**

Paper was clearly written and the general idea was easy to follow. There was a codebase available in PyTorch and part of the experiments were reproducible using this code.

**What was difficult**

The codebase was not clearly structured and has to be altered to produce results for most experiments reported in the paper. The reimplementation of the codebase was non-trivial due to otherwise undocumented details in the code having a large impact on outcomes.

**Communication with original authors**

The authors were contacted on multiple issues regarding implementation details and notation in the paper. Most of these were resolved swiftly and constructively. On two issues we remain in contact with the authors at this time.

Submitted to ML Reproducibility Challenge 2020. Do not distribute.

# 1  Introduction

Collaborative learning systems allow multiple users to jointly train a Deep Learning (DL) model. Each user has their own training data which is to calculate local gradients [18][7][12]. These local gradients are then shared among all users to update the parameters of the shared DL model, without the need of sensitive data leaving user's device. Primary benefit of federated learning is its capacity to improve the generalization of the resulting model, while maintaining privacy over the training data of individual users. This is especially important as confidentiality quickly becomes an essential quality of DL models [1]. Because of that, federated learning is used in applications from mobile networks [10] to autonomous driving [13] and health care [2].

However, this privacy benefit can be undone by reconstruction attacks as proposed by [6][19][20]. These attacks make it possible to reconstruct the original private training samples of users from the shared gradients of the federated learning system. This poses a considerable threat to the privacy of users of federated learning systems and the confidentiality of their data samples.

The paper subject to this reproducibility study proposes a novel approach to mitigate the threat from reconstruction attacks by augmenting the local training data of the user, before calculating the gradients [5]. Furthermore, the authors develop an automatic search algorithm to find the optimal transformation policies to augment the data and propose two novel metrics, $S_{pri}$ and $S_{acc}$, to increase the efficiency of this search.

In this reproducibility report, we evaluate the main claims made by the authors of [5] by reproducing their experiments. Moreover, we assess the availability of hyperparameters and other information needed for reproducibility, as well as discuss the usability of the provided codebase. We also extend the experimental setup towards a new dataset.

# 2  Scope of reproducibility

The main goal of the original paper is to develop an automatic search algorithm to find transformation policies that can defend privacy-sensitive training data against reconstruction attacks in a federated learning system. To achieve this, authors devise two novel metrics, described in Section 3.2. The main claims made in the paper are the following:

- **Claim 1**: by augmenting training samples with carefully-selected transformation policies, reconstruction attacks become infeasible
- **Claim 2**: the proposed search algorithm can find good and general policies, i.e. policies that are able to defeat multiple variants of reconstruction attacks
- **Claim 3**: the found policies are highly transferable; good policies searched for one dataset are also suitable for another datasets
- **Claim 4**: the found policies have negligible impact on the training efficiency
- **Claim 5**: in general, a good policy is made up of transformations that distort the details of the training samples, while maintaining the semantic information
- **Claim 6**: the five transformations that work best are *horizontal shifting (9)*, *brightness (9)*, *brightness (6)*, *contrast (7)* and *contrast (6)* (number inside the brackets represents the intensity of the applied transformation)
- **Claim 7**: $S_{pri}$ is a good measure of privacy; it is linearly correlated to Peak signal-to-noise ratio (PSNR) [9] with a Pearson Coefficient [15] of $0.697$

Each of these claims is supported by the results of one or more experiments in [5], represented in the tables and figures. In this reproducibility study, we rerun the experiments and reproduce the resulting tables and figures. In Section 5, we list which experiments support which claims. In Section 6, we discuss the reproducibility of each experiment and evaluate the validity of the claims.

Beyond reproducing the above claims from the original paper, we propose two extensions. Both of these extensions are based on the transferability of the searched policies as claimed in *Claim 3*. We test the transferability of the policies against additional dataset and evaluate whether the best performing transformations are the same on this dataset.

**Extension 1**: Using the policies searched on one dataset and applying them to a new dataset can make reconstruction attacks against this new dataset infeasible

**Extension 2**: Since good policies share the same general qualities, as claimed by *Claim 5*, the five best transformations from *Claim 6* are the same when using a different dataset.

In Section 5, we show the results for these extensions, and in Section 6, we relate them to the claims, experiments, and results from the original paper.

## 3   Finding privacy-preserving transformation policies

The original paper proposes an automatic search algorithm for finding privacy-preserving transformation policies. To better understand this main contribution, we take an in-depth look at what a transformation policy is and how good policies are found within a reasonable time.

### 3.1   Transformation policies

Transformations or augmentations have been widely used to improve model performance and generalizability in DL. In [5], transformations from AutoAugment[1] [3] are repurposed to protect sensitive training data from reconstruction attacks. The library contains 50 different transformations, including rotation, crop, shift, inversion, brightness, and contrast. A *transformation policy* is a combination of $k$ such transformations applied to the training samples. In [5], $k = 3$ is chosen and the policies are denoted by the indices of the transformations within the AutoAugment library.

Consistently apply the best policy to the data would risk domain shift in the dataset. Therefore, the authors propose the *hybrid strategy*, where a policy is randomly selected from the candidate policies - this way, good privacy and accuracy are guaranteed [5].

### 3.2   Reducing the search-space

To find candidate policies, it is necessary to determine their effect on both privacy and accuracy. The transformations must be applied to training data, and a model must be trained. Because fully training a model is very expensive, the authors propose two metrics that serve as a proxy for the privacy preservation and accuracy of the fully trained model: privacy-score($S_{pri}$) and accuracy-score($S_{acc}$). Low $S_{pri}$ entails the model has high privacy preservation potential, whereas high $S_{acc}$ means the model achieves good accuracy with the applied transformation policies. These metrics produce results on model that are trained with only 10% of the data for only 25% training iterations, reducing the search-space and making the policy search feasible in a reasonable time. Further details about the definition of $S_{pri}$ and $S_{acc}$ can be found in sections 4.2 and 4.3 of [5].

## 4   Experimental setup and code

To verify the claims made by the authors of [5], we reproduce their experiments. These experiments roughly fall into four categories: evaluating the effectiveness of the searched policies against reconstruction attacks, testing the transferability of the searched policies on different datasets and models, checking the impact on model efficiency, and studying the semantics behind the different transformations. Multiple models must be trained on augmented and un-augmented data for all these categories. For the attacks, the approach from [6] is applied. Section 5 provides a detailed description of the experiments and shows the results.

To reproduce the experiments performed by the authors, we used their existing codebase[2], which is implemented in PyTorch [14]. We refactored parts of this code and re-implemented the rest to our own version written in PyTorch Lightning[3], which leverages the interface advantages of the Lightning framework to make running experiments and logging results more intuitive. Main benefit of doing so is that more experiments can be tested with finer clarity and control of the setup. Our refactoring is this study's main contribution, and the codebase is publicly available at `https://anonymous.4open.science/r/MLRC2021-0454`.

---

[1]`https://github.com/DeepVoltaire/AutoAugment`
[2]`https://github.com/gaow0007/ATSPrivacy`
[3]`https://github.com/PyTorchLightning/pytorch-lightning`

### 4.1 Datasets

The experiments in [5] are performed on two datasets, CIFAR-100[4] [11], and Fashion-MNIST[5] [17]. CIFAR-100 contains $60,000$ color images of size $32 \times 32$, from 100 classes. The test set is used as the validation set, consistent with the authors' codebase. On the other hand, the Fashion-MNIST dataset contains $70,000$ grey-scale images of $28 \times 28$ resolution from 10 classes. Again the test set is used as a validation set. We run experiments on one additional dataset in our extensions - Tiny ImageNet200[6] [4]. It contains 120,000, $64 \times 64$ RGB images of 200 different classes. However, a *tiny* version of the dataset is introduced in the original paper for policy-search purposes. This dataset version contains $10\%$ of the original samples, using the same distribution. It's later used to train the models for the evaluation of $S_{pri}$ and $S_{acc}$ in the search algorithm.

### 4.2 Model descriptions

We use the following models:

- ResNet20-4, a variation of ResNet20 [8] that has four times the number of channels also used in [6]. The total number of parameters is 4.4M.
- ConvNet [6] - 8-layer Convolutional Neural Network, with batch normalization and a ReLU layer after each convolution layer. For this model the total number of parameters is 3.7M.

The original codebase uses the implementation of both models from the repository[7] of [6]. Our models are re-implemented in Pytorch Lightning. Both models were compared with the models from the original codebase in terms of accuracy; they achieved comparable results.

### 4.3 Hyperparameters

For policy search, we used $C_{max} = 1500$ and *max policies* equal to 10. The batch size was 128, and the number of transforms in policy was 3. For training, the batch size was also 128 and the number of epochs was 60 (see Section 4.4). To obtain a semi-trained network, we used a subset of 10% of the training dataset. The attack is performed on image with index 0, and we reused the remaining setups according to the original paper e.g. "inversed" (default attack). Except for Figure 4, where a default config was used, with a number of maximum iterations changed to 2500. For further experiments, we followed the same conventions.

### 4.4 Computational requirements

We ran our experiments using Nvidia GeForce GTX 1080 GPU. The policy search took approximately 10 hours. The training of one model took approximately 2h 40min using the original approach. However, training for 60 epochs achieves the same accuracy but in 50 minutes. It is because there exist periods of plateau, while lr is not scheduled yet to drop. One attack with 2500 iterations took approximately 5 minutes, so measuring the correlation between $S_{pri}$ and PSNR took 8.5 hours (with policy search).

## 5 Experiments and results

### 5.1 Results reproducing original paper

**Experiment 1** A reconstruction attack on 100 images from the CIFAR-100 validation set is performed with and without a searched transformation policy applied. We document the optimization process of the attack in terms of GradSim. The model used is ResNet20 trained on the tiny dataset for 50 epochs. The results of this experiment are shown in Figure 2, which shows a very similar result to the original paper. In addition to the original figure, we show the standard deviation over the 100 images, since GradSim can differ significantly from image to image. When taking the average of multiple runs, it can be seen that the privacy-aware transform does indeed make the GradSim convergence more difficult.

---

[4] https://www.cs.toronto.edu/~kriz/cifar.html

[5] https://github.com/zalandoresearch/fashion-mnist

[6] http://cs231n.stanford.edu/tiny-imagenet-200.zip

[7] https://github.com/JonasGeiping/invertinggradients

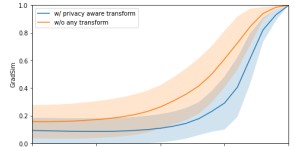
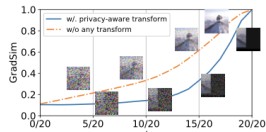
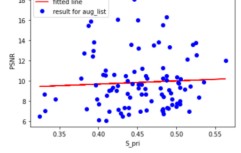
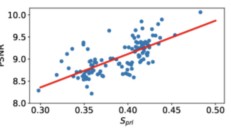

(a) Reproduced result    (b) Result from original paper

Figure 2: Optimization process of reconstruction attack with and without searched policy

(a) Reproduced results    (b) Original results

Figure 4: Correlation between $S_{pri}$ and PSNR

**Experiment 2**    A visual comparison between reconstructed images with and without a searched transformation policy applied is performed for both ResNet20 and ConvNet on images from CIFAR-100 and Fashion-MNIST. The optimizer used in the attack is Adam+Cosine. The images, the resulting reconstructions, and their PSNR values are shown in the left half of Figure 5. The results from the original paper are shown at the right side of Figure 5. As can be seen, the images used and PSNR values reported are different. This is due to the fact that it was too expensive to identify the exact same images and PSNR values differ quite severely depending on the image used. However, for all 12 images, we observe a less pronounced visual effect of the transformation policy as well as a smaller gap in PSNR values between the reconstructions with and without the policies applied. This implicates that the effect shown in the original paper is not as severe for all images, although the images we selected may be particularly easy to reconstruct.

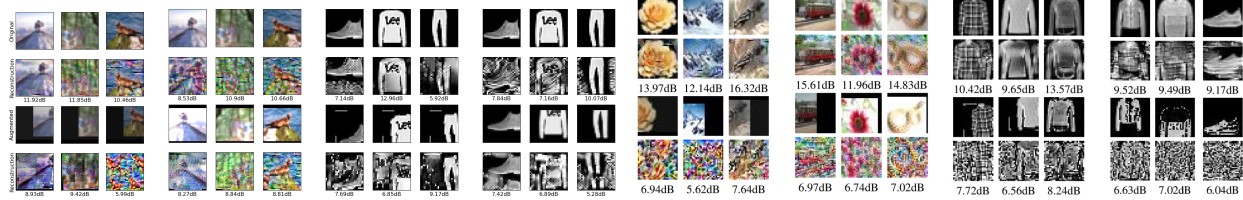

(a)  CIFAR-100(b)  CIFAR-100(c)      FMNIST(d)      FMNIST(e)  CIFAR-100(f)  CIFAR-100(g)      FMNIST(h)      FMNIST
with ResNet20  with ConvNet    with ResNet20  with ConvNet    with ResNet20  with ConvNet    with ResNet20  with ConvNet

Figure 5: Visualization results for reconstruction attacks on different datasets and models with associated PSNR values. Our results above and original results below.

**Experiment 3**    To gain further insight into the effectiveness of the different policies, we report the qualitative and quantitative results of Adam+Cosine attacks and model accuracy for the datasets and models in Figure 5. The results are calculated over 6 images as performing the experiment is very expensive and number wasn't stated in the paper. The policies considered and the results are listed in Table 1. Table 1 shows similar patterns to the original paper, where the

| Policy | PSNR | PSNR (std) | Acc |
|---|---|---|---|
| None | 12.15 | 2.06 | 78.11 |
| Random | 9.92 | 1.93 | 75.02 |
| 3-1-7 | 6.77 | 0.88 | 71.59 |
| 43-18-18 | 9.34 | 1.81 | 77.16 |
| Hybrid | 8.25 | 1.64 | 77.47 |

(a) CIFAR-100 + ResNet20

| Policy | PSNR | PSNR (std) | Acc |
|---|---|---|---|
| None | 11.44 | 2.93 | 72.97 |
| Random | 10.29 | 1.02 | 71.93 |
| 21-13-3 | 8.23 | 2.18 | 63.26 |
| 7-4-15 | 10.31 | 2.14 | 70.77 |
| Hybrid | 9.89 | 1.47 | 68.91 |

(b) CIFAR-100 + ConvNet

| Policy | PSNR | PSNR (std) | Acc |
|---|---|---|---|
| None | 9.81 | 4.41 | 95.19 |
| Random | 10.06 | 2.04 | 95.19 |
| 19-15-45 | 8.26 | 0.37 | 92.44 |
| 2-43-21 | 8.93 | 2.93 | 93.93 |
| Hybrid | 8.41 | 1.45 | 95.14 |

(c) FMINST + ResNet20

| Policy | PSNR | PSNR (std) | Acc |
|---|---|---|---|
| None | 9.52 | 3.27 | 94.61 |
| Random | 9.47 | 2.27 | 94.47 |
| 42-28-42 | 7.59 | 0.89 | 94.62 |
| 14-48-48 | 8.41 | 2.10 | 94.68 |
| Hybrid | 6.80 | 0.98 | 94.59 |

(d) FMINST + ConvNet

Table 1: PSNR (db) (including mean and standard deviation over 6 images) and model accuracy (%) of different transformation configurations for each model and dataset. $19 - 1 - 18$ is the random policy.

165 searched policies have low PSNR values compared to not using transformations. We do observe that PSNR values have
166 a relatively high standard deviation, and during our experiments, we found that the policies do not form a good defense
167 for some images. This problem will be further discussed in Section 6.

168 **Experiment 4** The defensive qualities of the searched transformation policies are benchmarked against existing
169 defenses from the literature [20] [16] under the Adam+Cosine attack. The results are shown in Table 6. Although the
170 exact values differ slightly, the overall results are similar to the original paper, where all the existing defenses perform
171 worse than the hybrid strategy.

172 **Experiment 5** This experiment concerns **Claim 2**. Because policies should be general, they are tested against various
173 attack configurations. For this, we again use 6 images from the test set and perform the different attacks on the images
174 without the transformation policies applied and with the hybrid strategy transformation policies applied. The results
175 are shown in Table 2. As can be seen from the table, the hybrid strategy works well against all configurations of the
reconstruction attack. This is in line with the results from the original paper.

| Attack | None | None (std) | Hybrid | Hybrid (std) |
|---|---|---|---|---|
| LBFGS+L2 | 8.61 | 1.22 | 6.33 | 2.00 |
| Adam+Cosine | 12.15 | 2.06 | 8.25 | 1.64 |
| LBFGS+Cosine | 9.62 | 0.91 | 7.47 | 0.25 |
| Adam+L1 | 9.48 | 0.71 | 6.43 | 0.16 |
| Adam+L2 | 9.28 | 0.69 | 6.46 | 0.21 |
| SGD+Cosine | 12.60 | 2.07 | 8.03 | 1.47 |

Table 2: PSNR values (db) (including mean and standard deviation over 6 images) of reconstructed images with and without transformations applied for different attack configurations

| Policy | PSNR | PSNR std |
|---|---|---|
| None | 15.39 | 2.78 |
| 3-1-7 | 8.47 | 0.85 |
| 43-18-18 | 10.97 | 1.06 |
| Hybrid | 8.95 | 0.90 |

Table 3: CIFAR100 with ResNet20

177 **Experiment 6** This experiment concerns the transferability of **Claim 3**. To test this, the policies searched on CIFAR-
178 100 are applied to Fashion-MNIST using both ResNet20 and ConvNet. Reconstruction attacks are performed with
179 the Adam+Cosine attack. The resulting PSNR values and accuracies are listed in Table 4. The results differ from the
original. It can be seen that the transformation policies are not effective here.

| Policy | PSNR | PSNR (std) | Acc |
|---|---|---|---|
| None | 9.81 | 4.41 | 95.19 |
| 3-1-7 | 9.30 | 2.72 | 93.20 |
| 43-18-18 | 10.03 | 2.23 | 94.88 |
| Hybrid | 7.49 | 1.57 | 94.49 |

(a) FMNIST + ResNet20

| Policy | PSNR | PSNR (std) | Acc |
|---|---|---|---|
| None | 9.52 | 3.27 | 94.61 |
| 21-13-3 | 9.99 | 2.12 | 92.38 |
| 7-4-15 | 9.34 | 1.62 | 94.35 |
| Hybrid | 11.50 | 5.80 | 93.77 |

(b) FMNIST + ConvNet

180 Table 4: Resulting PSNR (dB) and accuracy (%) values for applying policies searched on CIFAR-100 to Fashion-MINST

181 **Experiment 7** The following experiment is aimed at **Claim 4**. The authors state that applying the search policies has
182 a negligible impact on training efficiency. We trained ResNet20 with the searched policies applied and documented the
183 loss and accuracy convergence to test this. From Figure 6 it can be seen that indeed applying transformations has almost
184 zero impact on the training efficiency. It is also noteworthy to observe that the training curves are almost identical
compared with the results from the original work.

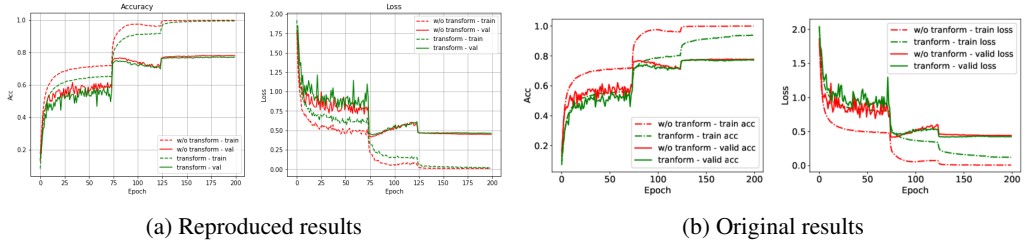

(a) Reproduced results        (b) Original results

185 Figure 6: Convergence speed with and without transformations applied

**Experiment 8**    **Claim 5** states that good transformation policies obfuscate details in the training samples but maintain high-order semantic information. As such, attackers will have trouble reconstructing high frequency information. We test this by comparing the attacker-defender gradient similarity during an attack of models trained with the searched policy, a random policy, and no policy applied. From Figure 7, it can be seen that in shallow layers, the gradients differ significantly, whereas in deep layers, the gradients are very similar. This implies that the transformations do indeed have the desired effect and is in line with the results from the original paper.

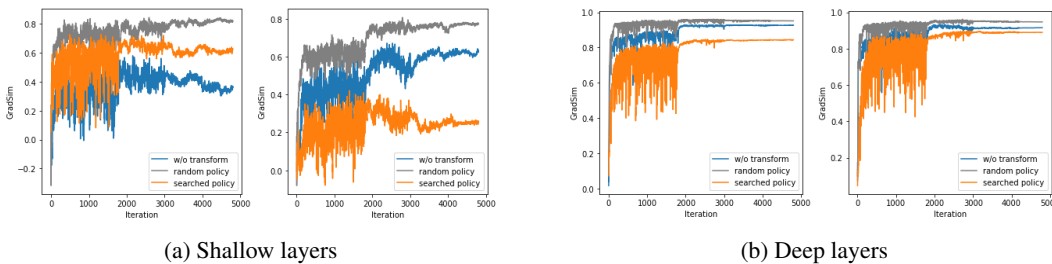

(a) Shallow layers                                    (b) Deep layers

Figure 7: Reproduced results of gradient similarity during the reconstruction optimization, for CIFAR100 with ResNet20

**Experiment 9**    In **Claim 6** the authors report their 5 top transformations. We test whether we can find the same ones by calculating the privacy score on the dataset for each individual augmentation and show the results in Figure 8a and 8b. Out of the best 5 transformations reported in the original paper we found 4 overlapping ones.

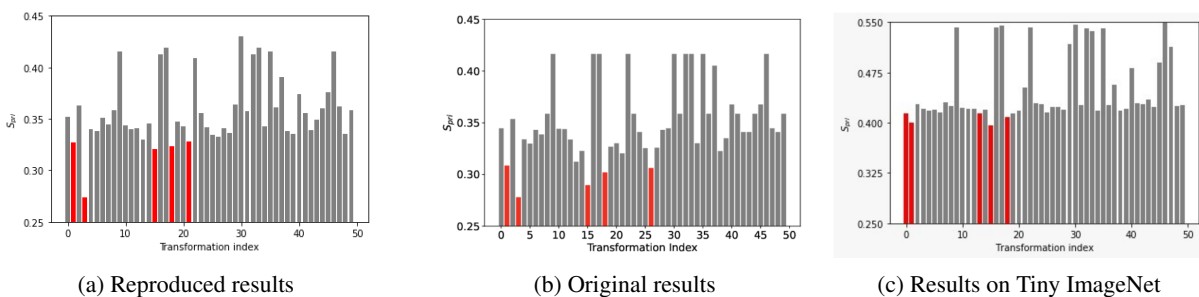

(a) Reproduced results                 (b) Original results                 (c) Results on Tiny ImageNet

Figure 8: Privacy scores of the 50 transformation functions in the augmentation library, best transformations are red.

**Experiment 10**    The final experiment reproducing the results from the original paper is aimed at **Claim 7**. The authors claim that their privacy-score $S_{pri}$ is linearly correlated with PSNR with a Pearson-coefficient of $0.697$. We test this by running attacks and evaluating $S_{pri}$ on the model trained on tiny cifar100 for 50 epochs and found a very different result. As shown in Figure 4 there is hardly any correlation (Pearson-coefficient is $0.123$). This might be due to the fact that these $100$ transformation policies are selected at random out of $127.550$ possible options. This is a striking result nonetheless, which we discuss in-depth in Section 6.

## 5.2    Results beyond original paper

**Extension 1**    We extend the evaluation of the transferability of the searched policies by evaluating the performance of the policy searched on CIFAR-100 on Rescaled ImageNet. The resulting PSNR values and accuracies are shown in Table 5. As can be seen from the table, the hybrid strategy produces only $1$ dB improvement in PSNR value, and accuracy decreases by more than $4\%$. This weakens the claim of transferability made by the authors.

**Extension 2**    We additionally extend the evaluation of the transferability of the searched policies by testing which transformations work best on a different dataset. Since good policies share the same general qualities, as stated in Claim 5, the five best transformations from Claim 6 can be expected to be the same when using a different dataset. For this experiment, we use the Rescaled ImageNet dataset. The resulting transformations are shown in Figure 8c. Out of the $5$ best transformations on the Rescaled ImageNet $3$ were also found on CIFAR-100 in both our results and the results from the original paper. This shows that, indeed, these transformations contain the desired qualities from Claim 6.

| Policy | PSNR | PSNR (std) | Acc |
|--------|------|-----------|-----|
| None | 8.96 | 1.25 | 61.44 |
| Hybrid | 7.92 | 0.79 | 57.38 |

Table 5: PSNR values (dB) and accuracies of policies searched on CIFAR-100 applied to Rescaled ImageNet

| Defense | PSNR | PSNR (std) | Acc |
|---------|------|-----------|-----|
| Pruning (70%) | 11.62 | 2.18 | 74.61 |
| Pruning (95%) | 10.41 | 1.32 | 67.91 |
| Pruning (99%) | 9.96 | 0.57 | 53.43 |
| Laplacian ($10^{-3}$) | 10.73 | 1.02 | 71.45 |
| Laplacian ($10^{-2}$) | 12.03 | 0.79 | 26.20 |
| Gaussian ($10^{-3}$) | 12.11 | 2.98 | 72.89 |
| Gaussian ($10^{-2}$) | 12.13 | 1.14 | 36.25 |

Table 6: Comparisons with existing defense methods under the Adam+Cosine attack

# 6 Discussion

Overall the results in [5] are reproducible, except Figure 4, with a large discrepancy between our result and the original one - we are still in contact with the authors on this issue. Nevertheless, augmentation policies tend to work as a defense mechanism rather well. For most images, an attacker using reconstruction attacks is unable to find privacy-sensitive information. However, the standard deviation of our results is more than $25\%$ in some settings, and we consider this a valuable metric to contribute. Some images are vulnerable to the attack even with the proposed defense mechanism, and it is as of yet unclear to us which types of images are more vulnerable than others. This issue must be developed further in future research to make the approach widely applicable in real-world use-cases where private data is at stake.

Additionally, we made observations in the codebase that, to the best of our knowledge, were not reported in the paper or any other accompanying documentation. The first was the fact that the loss of the training module was multiplied by a factor of $0.5$. This is not a fundamental flaw during the training phase, as it simply produces smaller gradients and therefore leads to a reduced effective learning rate. However, during the reconstruction attacks, the loss used by the attacker was not multiplied by this factor. This makes the attacker in practice use a different loss function from the one used to generate the gradient that it is attempting to match. This may therefore make reconstruction more difficult. Furthermore, we found that two other undocumented augmentations were added in all experiments, namely a random crop and random horizontal flip. Without these, the accuracy of our models decreased by over $10\%$. We are in contact with the authors regarding these observations, they acknowledged the halved loss as a bug.

## 6.1 What was easy

The explanation of the general idea and solution of the paper was very clearly put and easy to follow. The codebase contained a README with instructions on how to run some of the paper's experiments, and these instructions could be followed without significant problems. The code produced results as seen in the paper.

## 6.2 What was difficult

The most challenging part about reproduction was the unclear description of experiments in the paper and limited clarity in the codebase. Code in the repository was uncommented, used many global variables and many layers of indirection. Many chunks of code were not used, making it harder to follow. Some experimental settings and metrics were not implemented, and some experiment configurations led to fatal errors.

It was very unclear which steps were originally followed to obtain Figure 4. Despite the authors' helpful comment on which model was used, we were not able to reproduce the correlation, potentially due to randomness in a vast search space (127,550) and the limited sample size (100). Furthermore, the paper does not state how many images were used to produce the PSNR values in the tables. Finally, undocumented augmentations were added in some but not all settings, which was cause for some delay until this was found to be the cause for a 10% accuracy-gap with the authors' results.

## 6.3 Communication with original authors

We contacted the authors about multiple clarifications regarding implementation details and notation in the paper. The authors responded promptly and answered almost all of our questions in the first round of contact. We are still in contact on two points. Firstly, regarding our reproduction of Figure 4. Since we got such differing results for this critical part of the authors' work, we are looking to investigate this further and possibly resolve the discrepancy with them. Secondly, we offered our refactoring of the codebase to the authors as a contribution to their work.

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
