# OpenReview forum: "Replication study of "Privacy-preserving Collaborative Learning with Automatic Transformation Search""
_ML_Reproducibility_Challenge/2021/Fall — RC2021_

### Official Review · Reviewer_FPrB · 2022-03-01
**Reprodicibiliy for "Privacy-preserving Collaborative Learning with Automatic Transformation Search"**

**Rating:** 7
**Confidence:** 4

**Review:**

This submission is based on reproducing and evaluating claims made in the paper - Privacy-preserving Collaborative Learning with Automatic Transformation Search. The document is well written and clearly explains not only the motivation and work done in the original paper being replicated, but also clearly mentions the efforts and work done in the reproducibility experiments. The authors have done a great job in clearly stating the the different claims made by the original paper, the motivation behind each of them, and the approach they took to try to validate their claims.

Some of the other positive things in this submission are -
1. All the main claims in the original paper were tested and the results properly presented in the document.
2. Extension of the datasets by adding an additional dataset, which is especially useful to validate the claims related to transferability and goodness of the proposed policies.
3. Separate sections of results for each claim from the original paper.
4. Re-implementation and extension of the existing codebase to PyTorch-lightning, which would potentially make it easier for other researchers to use the codebase and extend the work.
5. Clearly indicating parts which could not be replicated, and the efforts made by the authors to get in touch with the original authors about clearing up the mismatched results, a couple of which are an ongoing discussion.

A few things which could help authors to further make their contribution useful to even broader community:
1. Update their document and codebase's readme once the discrepancies with the original paper's authors have been resolved. Since this is still an ongoing issue, updating it in the final version would help future re-implementations and any references to the results.
2. Extend the captions of the images and tables in the experiments and results section to have a quick reference for the reader on which result was perfectly reproduced and which one had some discrepancies.
3. Update and extend the ReadMe of their codebase link with the main contributions and highlights to give a quick overview to the user on what worked and what couldn't be replicated, with a brief mention of any ongoing discussions on discrepancies.

Overall, the authors have made a good contribution to the paper, and the improvement of the codebase is also a very useful addition.

---

### Official Review · Reviewer_KC8W · 2022-03-01
**Re:Replication study of "Privacy-preserving Collaborative Learning with Automatic Transformation Search"**

**Rating:** 7
**Confidence:** 4

**Review:**

•	Reproducibility Summary: Yes
The replication study using structured section as their producibility summary in the first page, specifically it begins with Scope of Reproducibility

•	Scope of reproducibility
The authors attempted to test all the main claims made by the authors of the paper to reproduce. In addition, the authors extended work to a new dataset.

•	Code
They used an existing codebase at https://anonymous.4open.science/r/MLRC2021-045

•	Communication with original authors: Yes
They contacted the authors for multiple clarifications regarding implementation details and notations in the paper.

•	Hyperparameter search: No
The hyperparameter search was performed with extending the work in the original paper to try new settings.

•	Ablation Study: Yes
They conducted “search-space reducing” to shorten the computational time, which can be partly regarded as an ablation study.


•	Discussion on results: Yes
Detailed Discussion is provided, comparing their results with those from the original paper. Two separate sections were devoted to the discussion of what was easy and what was difficult.

•	Recommendations for reproducibility: provided
Results different from those in the original paper were reported.

•	Results beyond the paper: Yes
The replication study used two additional datasets: Rescaled ImageNet3 and CIFAR-100.

•	Overall organization and clarity
This report is well organized and written, easy to follow. More in depth discussion could strength this reproducibility study.

---

### Official Review · Reviewer_MgzW · 2022-03-06
**An excellent work**

**Rating:** 9
**Confidence:** 4

**Review:**

The research presented a replication of privacy-preserving collaborative learning with automatic transformation search. The authors were able to test the claims made by the research by rerunning and reporting the result that was in line with the original result of the experiments. The evaluation of the reproducibility of the paper was done through constant communication with the original authors.

The authors did a good work, they extended the previous research by testing the model using new dataset.

---

### Meta-Review · Program_Chairs · 2022-04-09

**Recommendation:** Accept
**Confidence:** 5

**Metareview:**

A solid contribution to the reproducibility challenge.  The submission is accepted.

---

### Decision · Program_Chairs · 2022-04-09

**Decision:**

Accept

**Comment:**

Following the recommendation of reviewers and meta-reviewer, the paper is accepted for ML Reproducibility Challenge 2021, and will be published in the upcoming special edition of ReScience Journal.